# Dynamics and Half-Life of Cell-Free DNA After Exercise: Insights from a Fragment Size-Specific Measurement Approach

**DOI:** 10.3390/diagnostics15010109

**Published:** 2025-01-04

**Authors:** Ryutaro Yamamoto, Hiroshi Asano, Ryo Tamaki, Yoshihiro Saito, Ami Hosokawa, Hidemichi Watari, Takeshi Umazume

**Affiliations:** Department of Obstetrics, Hokkaido University Hospital, Sapporo 060-8648, Japan; yamaminto@pop.med.hokudai.ac.jp (R.Y.);

**Keywords:** cell-free DNA, biomarker, circulating, half-life, clearance, fragment size

## Abstract

**Background:** Cell-free DNA (cfDNA) is present in healthy individuals but is elevated in those undergoing physical exertion, trauma, sepsis, and certain cancers. Maintaining cfDNA concentrations is vital for immune homeostasis and preventing inflammatory responses. Understanding cfDNA release and clearance is essential for using cfDNA as a biomarker in clinical diagnostics. We focused on the fragment size of cfDNA and investigated cfDNA dynamics and half-life, particularly the 100–250 base pair fragments. **Methods**: Healthy, adult men (*n* = 5; age 40 ± 4.1 years) were subjected to a 30 min treadmill exercise. Blood samples were collected at 0, 5, 10, 15, 30, and 60 min post-exercise using PAXgene^®^ Blood ccfDNA tubes to stabilize and prevent nuclease-mediated cfDNA degradation and minimize genomic DNA contamination risk. The cfDNA concentration was measured using an electrophoresis-based technique (4150 TapeStation system) to quantify the concentration based on cfDNA fragment size. **Results**: The results showed a cfDNA half-life of 24.2 min, with a transient increase in 100–250 base pair cfDNA fragments post-exercise, likely due to nuclease activity. These levels rapidly reverted to the baseline within an hour. **Conclusions**: The rapid clearance of cfDNA underscores its potential as a biomarker for real-time disease monitoring and the evaluation of treatment efficacy. This study is expected to standardize cfDNA investigations, enhancing diagnosis and treatment monitoring across various disease conditions.

## 1. Introduction

Cell-free DNA (cfDNA), which refers to DNA fragments that circulate freely in the cytoplasm outside of cells, was first identified in 1948 [1]. Primarily composed of short DNA fragments, typically 160–180 base pairs (bp) [2,3], cfDNA released into the bloodstream can originate from various processes such as apoptosis, cell death, and tumor destruction [4,5]. The discovery of fetal cfDNA in the plasma of pregnant women revolutionized prenatal care, enabling the development of noninvasive prenatal testing (NIPT), which analyzes fetal DNA fragments in the maternal blood [6,7,8]. Since cfDNA plays a significant role in various physiological and pathological processes, researchers are actively exploring the potential of plasma cfDNA as a biomarker, with its application in cancer diagnosis, prognosis, and treatment monitoring—often referred to as liquid biopsy—being among the most promising [9,10]. Recent studies have increasingly explored the potential of cfDNA as a biomarker not only in oncology but also in autoimmune diseases, traumatic conditions, sepsis, cardiovascular diseases, and sports medicine [11,12,13,14,15].

The metabolism and clearance of cfDNA primarily occur in the liver, kidneys, and spleen, with its removal facilitated by degradation through nucleases, phagocytosis, and immune complex formation [16,17]. In particular, Kupffer cells in the liver play a key role in removing long DNA fragments [18]. The kidneys also contribute by breaking down DNA fragments through deoxyribonuclease activity [17]. Under normal physiological conditions, the body efficiently eliminates cfDNA, helping to maintain its concentration within controlled limits [19]. However, research on the clearance rate and half-life of cfDNA is still evolving, and a comprehensive understanding of how long cfDNA remains in circulation is essential for its effective utilization as a biomarker in clinical settings. Four small-scale studies have investigated the half-life of cfDNA, with estimates ranging from approximately 15 min to 2 h [20,21,22,23]. In the study by Lo et al. [20], cfDNA was extracted from blood collected in ethylenediaminetetraacetic acid (EDTA) tubes over time from eight women who had given birth to male fetuses, and the cfDNA concentration was measured using real-time quantitative polymerase chain reaction (PCR) targeting the sex-determining region Y (*SRY*) gene from the fetuses. Following delivery, the concentration of the *SRY* gene gradually decreased, with an average half-life of 16.3 (range 4–30) min, reflecting the time required for a 50% reduction in concentration [20]. Other studies have reported cfDNA half-lives of about 15 to 114 min by applying various methods [21,22,23].

However, these studies had several limitations. First, they lacked measures to prevent cfDNA degradation by nucleases in the blood, particularly degradation caused by nuclease activity in the blood after collection. Second, they depended on PCR-based methodologies, which can introduce biases by amplifying both cfDNA and genomic DNA (gDNA) without discrimination. Third, fragment size may also influence the half-life of cfDNA, but no studies have yet specifically focused on its dynamics by fragment size [24].

To address these limitations, we designed this study to clarify the dynamics and half-life of cfDNA following exercise, employing an improved, fragment size-specific approach. Specifically, we used blood collection tubes designed to inhibit nuclease activity, employed an electrophoresis-based technique (4150 TapeStation system) to measure cfDNA concentration based on fragment size instead of PCR, and focused exclusively on measuring the concentration of 100–250 bp fragments. The aim of this exploratory investigation was to evaluate the clearance of cfDNA in the blood following an exercise-induced increase.

## 2. Materials and Methods

### 2.1. Study Design and Ethical Considerations 

Five healthy adult participants working at Hokkaido University Hospital were subjected to a mild treadmill exercise regimen lasting 30 min, with blood samples collected at multiple time points to analyze the dynamics of cfDNA and its half-life. Because of the invasive nature of the study, which entailed collecting blood samples seven times, the sample size was limited to five participants. 

This study was reviewed and approved by the Ethics Committee of the Faculty of Medicine at Hokkaido University (Approval Number: 023-0064) and adhered to the principles outlined in the Declaration of Helsinki. Informed consent was obtained from all participants prior to blood sampling, including consent for participation and publication of the study results.

### 2.2. Participants and Eligibility

Men aged 18 years or older who had no chronic medical conditions prior to the exercise stress test, were fully informed of the study aims and procedures, and voluntarily consented to participate were included in the study. Exclusion criteria were chronic liver, kidney, or thyroid disease, or the regular intake of oral medication. 

The average age of the five study participants (men) who met the inclusion criteria was 40 ± 4.1 years, with a height of 169 ± 7.3 cm, weight of 70 ± 15 kg, and body mass index (BMI) of 24.4 ± 4.7 kg/m² (Table 1). All participants were in good health, with no chronic diseases or any regular medication use. None of the participants had regular exercise habits, nor did they report any musculoskeletal pain on the day of the test. All of them successfully completed the 30-min treadmill exercise as scheduled.

### 2.3. Exercise Regimen and Sample Collection

On the day of the test, after confirming that the participants were in good physical condition, blood samples were collected while they were in a relaxed sitting position. Following this, the participants engaged in a 30 min treadmill exercise at a controlled speed of 8 km/h, which was set and maintained on the treadmill to ensure consistency and reproducibility across all participants.

Blood samples were taken immediately after completing the exercise regimen, and at 5, 10, 15, 30, and 60 min thereafter. All samples were collected while the participants remained in a resting sitting position, and they were instructed to maintain this position until all blood sample collections were completed. Blood was drawn from the median basilic vein or cephalic vein in the forearm, considered optimal sites for blood collection. PAXgene^®^ Blood ccfDNA tubes (QIAGEN, Hilden, North Rhine-Westphalia, Germany) were used for collection, with 10 mL of blood collected per tube. All samples were stored at 4 °C, and cfDNA was isolated 1–3 days after blood collection.

### 2.4. Isolation of cfDNA

The blood samples were centrifuged at 4 °C for 15 min at 1900× *g*. The plasma supernatant was carefully aspirated to avoid disturbing the separation layers, and 4–5 mL of plasma was transferred into polypropylene 15 mL conical tubes (catalog no. 352096; Falcon, Corning, NY, USA). A second centrifugation was performed at 4 °C for 15 min at 1900× *g*. The plasma supernatant was then carefully aspirated again, and approximately 4 mL of plasma was aliquoted into new 15 mL conical tubes.

The extraction of cfDNA was performed using the QuickGene cfDNA isolation kit (catalog no. CF-L; KURABO, Osaka, Japan) as the extraction reagent and the QuickGene-Mini8L (KURABO, Osaka, Japan) as the extraction device, following the manufacturer’s instructions. Initially, 600 μL of a protease mixture was dispensed into a polypropylene 50 mL conical tube (catalog no. 352070; Falcon, Corning, NY, USA), followed by the addition of 4 mL of plasma. An amount of 5 mL of Lysis Buffer was added within 30 s to break down unwanted proteins and stabilize the cfDNA, and the mixture was vigorously shaken up and down for 10 s. After vortexing for 30 s at a maximum speed of 2500 rotations per minute (rpm), the sample was incubated in a water bath at 56 °C for 5 min. Subsequently, 2.4 mL of special-grade ethanol (>99%) was added, and within 30 s, the mixture was again vigorously shaken up and down for 10 s, followed by another 30 s vortex at the same speed (2500 rpm).

The sample was injected into the filtered cartridge of the QuickGene-Mini8L. After allowing a standing time of 10 min, or until the entire volume had eluted, additional pressure was applied to recover the remaining volume. Next, to wash away unwanted components adhered to the cartridge, 7.5 mL of Wash Buffer was added, left to stand for 5 min, and then pressurized. This process was repeated with 6.5 mL and 5.5 mL of Wash Buffer, each time allowing the mixture to stand for 5 min before applying pressure.

For DNA recovery, the cartridge holder was moved to the elution position. A total of 0.1 mL of Elution Buffer was applied to the filter, allowed to stand for 3 min, and pressurized. The recovered cfDNA was collected into a 1.5 mL microcentrifuge tube. Finally, the cfDNA was stored at −20 °C for long-term preservation.

### 2.5. Measurement of cfDNA and Data Analyses

The concentration of cfDNA was measured using the 4150 TapeStation system (Agilent Technologies, Santa Clara, CA, USA). A 1:1 mixture of 2 μL extracted cfDNA and 2 μL High-Sensitivity D1000 Sample Buffer was prepared in an 8-tube strip for analysis. Additionally, 2 μL of High-Sensitivity D1000 Ladder was prepared with the Sample Buffer and loaded into position A1 of the tube strip holder. The High-Sensitivity D1000 ScreenTape, the filter-loaded tip, and the prepared sample were loaded into the 4150 TapeStation system for analysis.

The intra-rater reliability was assessed using the Intraclass Correlation Coefficient (ICC). ICC (1,1) was calculated for the single rater using the absolute agreement, one-way random effects model. Python (version 3.11.8) was utilized for the analysis.

The data obtained from the TapeStation system were analyzed using TapeStation Software version 5.1 (Agilent Technologies, Santa Clara, CA, USA). To effectively visualize the results, GraphPad Prism version 10.4.0 (GraphPad Software, San Diego, CA, USA) was used to generate graphs. DNA fragment sizes were represented by red lines for 100–250 bp fragments and blue lines for 400–1200 bp fragments. Time progression was shown as a line graph.

### 2.6. Literature Search

To compare the results of this study with previous reports, a systematic literature search was conducted. The search included PubMed, Google Scholar, and the Cochrane Library, targeting relevant studies published between 1 January 1999 and November 2024. The keywords “Cell-free DNA (cfDNA)”, “Circulating tumor DNA (ctDNA)”, “Half-Life”, and “Clearance” were used in various combinations, focusing on studies where “Half-Life” or “Clearance” appeared in the title or abstract. The search was limited to articles focusing on circulating cfDNA in the blood, with a restriction to human subjects. The characteristics of study participants, blood collection tubes used, measurement methods used, and data on cfDNA half-life were organized in tabular form and compared across studies.

## 3. Results

### 3.1. Quantitative and Qualitative Analysis of cfDNA

Blood samples were collected at 0, 5, 10, 15, 30, and 60 min post-exercise, and the concentration and size of cfDNA were measured and analyzed using the 4150 TapeStation system. The analysis yielded an ICC (1,1) value of 0.943 (95% CI: 0.501–0.919), indicating good reliability. Figure 1 illustrates the measurement outcomes for participant 2 taken 15 min after exercise. The cfDNA displayed two distinct peaks: 100–250 bp and 400–1200 bp. The concentration of 100–250 bp fragments was higher than that of 400–1200 bp fragments, a trend observed in all participants. The most frequently detected fragment size ranged from 140 to 180 bp.

### 3.2. Post-Exercise Changes in cfDNA Concentrations

Figure 2 illustrates the time course of changes in cfDNA concentration before and after exercise of each of the study participants, represented by a line graph. Following exercise, the concentration of cfDNA in plasma increased immediately and subsequently declined. The concentration of 100–250 bp fragments consistently exceeded that of 400–1200 bp fragments. Notably, the concentration of 100–250 bp fragments did not exhibit a steady decline; transient increases were observed in cases 1, 2, 3, and 4. This phenomenon was believed to be caused by the breakdown of 400–1200 bp fragments by nucleases, resulting in a brief spike in the concentration of 100–250 bp fragments.

### 3.3. Analysis of 100–250 bp and 400–1200 bp cfDNA Half-Life

Table 2 presents data only pertaining to the concentration of 100–250 bp cfDNA, including baseline levels, maximum values, and half-life. Prior to exercise, the concentration of 100–250 bp cfDNA was low, at 91.7 (22.3–544) pg/μL. After exercise, however, the highest concentration more than doubled in all participants, rising to 1870 (738–3760) pg/µL, though the rate of rise varied between individuals. The average half-life after exercise was determined to be 24.2 (14.9–32.7) min. The amount of cfDNA returned to levels close to the initial values 60 min after activity in all cases. Similarly, the data specific to 400–1200 bp cfDNA showed an average half-life of 23.3 (9.5–50.8) min. The results indicated that the average half-life of 400–1200 bp cfDNA was slightly shorter than that of 100–250 bp cfDNA.

### 3.4. Half-Life of Cell-Free DNA in Literature Search

A systematic literature search using PubMed, Google Scholar, and the Cochrane Library identified a total of 15 studies. Of these, six articles were excluded because they were review articles. Among the remaining nine studies, four directly evaluated the half-life of cfDNA (Table 3). The study populations included pregnant women, cancer patients, and healthy adults, and all studies utilized EDTA tubes for blood collection. PCR was the primary measurement method, with reported cfDNA half-lives ranging widely from 15 to 114 min.

Lo et al. [20] (1999) analyzed the clearance of fetal-derived cfDNA in maternal blood among pregnant women. This study reported that fetal cfDNA concentrations decreased rapidly after delivery, with an average half-life of 16.3 min (range: 4–30 min), measured using PCR targeting the SRY gene.Diehl et al. [21] (2008) investigated the dynamics of circulating tumor DNA (ctDNA) in cancer patients. Using real-time PCR, the study measured changes in ctDNA concentrations in the blood of a single patient after tumor resection, reporting a half-life of 114 min.Breitbach et al. [22] (2014) examined changes in cfDNA concentrations in healthy adults after exercise. Blood samples were collected using small EDTA-coated capillary tubes, and real-time PCR was employed for measurement. While the average half-life was 15 min in under 50% of cases, cfDNA displayed varied kinetics in other cases, making a consistent evaluation of half-life challenging.Yao et al. [23] (2016) studied healthy adults, assessing the degradation rates of both naked DNA and DNA–protein complexes in serum samples. The half-life of naked DNA was 30.8 min, whereas DNA bound to proteins exhibited a much longer half-life of 157.6 min.

## 4. Discussion

In this study, we focused on the fragment size of cfDNA, examining its concentration and clearance, with particular attention paid to the 100–250 bp fragments. Two significant findings emerged from our investigation. First, concentrations of 100–250 bp cfDNA increased post-exercise and then rapidly declined, with a half-life of 24.2 min; it was confirmed that 100–250 bp cfDNA concentrations reverted to pre-exercise levels within one hour after the end of exercise. Second, a transient increase in levels of 100–250 bp cfDNA fragments was observed 15–30 min post-exercise cessation. This phenomenon was believed to result from the degradation of 400–1200 bp cfDNA fragments by nucleases, leading to the generation of additional 100–250 bp fragments.

Our findings align with previous studies on cfDNA dynamics, which have reported an increase in cfDNA levels following exercise [22,25,26,27,28]. While studies directly examining the half-life of cfDNA are relatively scarce, existing research suggests that cfDNA half-life typically ranges from 15 to 114 min. Although variations in study populations and methodologies complicate direct comparisons, our finding of a 24.2 min half-life is consistent with these reported values and supports their validity. Importantly, prior research has not investigated cfDNA clearance based on fragment size. To the best of our knowledge, this study is the first to explore cfDNA clearance specifically categorized by fragment size.

In this study, several strategies were employed to conduct an analysis specifically focused on the fragment size of cfDNA. The use of PAXgene^®^ Blood ccfDNA tubes, which contain anticoagulants, effectively stabilized cfDNA, protecting it from degradation by nucleases, and minimized the risk of gDNA contamination from leukocytes [29]. This stabilization not only preserved the integrity of circulating cfDNA but also allowed for the precise measurement of fragment size [30,31,32,33]. We also utilized the 4150 TapeStation system to measure cfDNA concentration, which employs electrophoresis to quantify DNA fragments based on their size by minimizing the risk of gDNA contamination, offering a distinct advantage over conventional PCR methods [34]. In addition, while PCR is an effective method for amplifying specific gene sequences, it can cause incorrect measurements of gDNA and other nonspecific DNA, since PCR may amplify all of the DNA in a sample, including impurities like gDNA and other undesired DNA fragments. In contrast, the 4150 TapeStation System measures DNA concentrations based on the physical size of DNA fragments, facilitating an accurate analysis of cfDNA concentration and size [35]. This approach enables a more precise understanding of cfDNA dynamics and allows for a focused analysis of fragments of specific sizes such as 100–250 bp.

The observed transient increase in 100–250 bp fragments following exercise is crucial for a comprehensive understanding of the cfDNA degradation process. The concentration of 100–250 bp fragments appeared to result from the breakdown of the 400–1200 bp cfDNA fragments into smaller sizes. This observation suggests that breakdown products from gDNA may have been inadvertently included in conventional procedures, perhaps resulting in inaccuracies in previous studies on the half-life of cfDNA. Further validation is required to confirm this mechanism.

While the half-life of 24.2 min observed in this study falls within the range reported in previous studies (15–114 min), it is longer than the 15 min half-lives reported in some studies [20,22]. We believe that the fragment size-specific measurement approach allowed us to capture cfDNA dynamics with greater precision. The utilization of PAXgene^®^ Blood ccfDNA tubes significantly mitigated gDNA contamination, while the precise size-specific quantitative measurement achieved with the 4150 TapeStation system contributed to the reliability of the data obtained. Furthermore, by focusing the analysis on 100–250 bp fragments, we were able to evaluate the clearance rate of cfDNA with greater accuracy.

A limitation of this study is the small sample size of only five participants due to the invasive nature of the study. Nevertheless, consistent results were observed for all of them, which strengthens the reliability of the findings. Consequently, the outcomes of this study exhibit a degree of validity and can provide an initial foundation for future large-scale investigations. Additionally, since the study participants were exclusively male, variations related to sex were not assessed. Previous studies have also reported variability in cfDNA half-life, and consistent with these findings, our study observed inter-individual differences. This suggests the need to evaluate factors such as age, renal and hepatic function, muscle mass, and nuclease activity to better understand cfDNA clearance mechanisms. Future research needs to involve larger sample sizes and explore cfDNA dynamics across various physiological and pathological conditions to address these factors comprehensively.

## 5. Conclusions

Our study determined that the half-life of 100–250 bp cfDNA fragments is approximately 24.2 min, indicating rapid clearance. This rapid turnover highlights the suitability of cfDNA as a dynamic biomarker for monitoring disease progression and treatment response. This is the first study to specifically calculate the half-life of 100–250 bp cfDNA fragments using the PAXgene^®^ Blood ccfDNA tubes and 4150 TapeStation system. The findings of this research will provide a scientific foundation for enhancing the clinical utility of cfDNA.

## Figures and Tables

**Figure 1 diagnostics-15-00109-f001:**
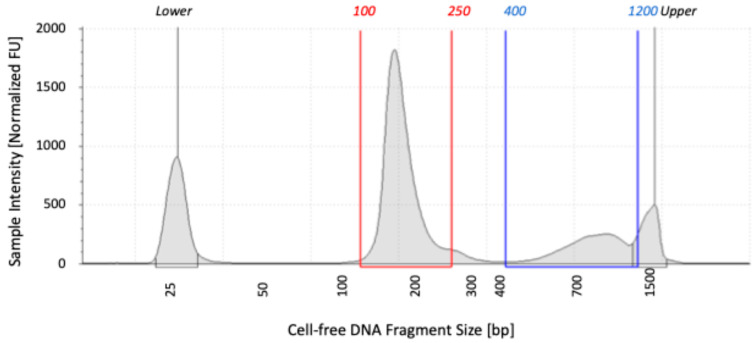
Quantitative and qualitative analysis of cell-free DNA (cfDNA) obtained from a blood sample extracted from participant 2 15 min post-exercise using 4150 TapeStation system (Agilent Technologies, Santa Clara, CA, USA). The vertical axis represents the fluorescence intensity of the sample, while the horizontal axis indicates the size in base pairs (bp) of the DNA fragments. The upper and lower markers delineate the minimum and maximum size ranges of the sample, respectively. The primary peak of cfDNA falls within the range of 100–250 bp, highlighting the predominance of fragments approximately 170 bp in length.

**Figure 2 diagnostics-15-00109-f002:**
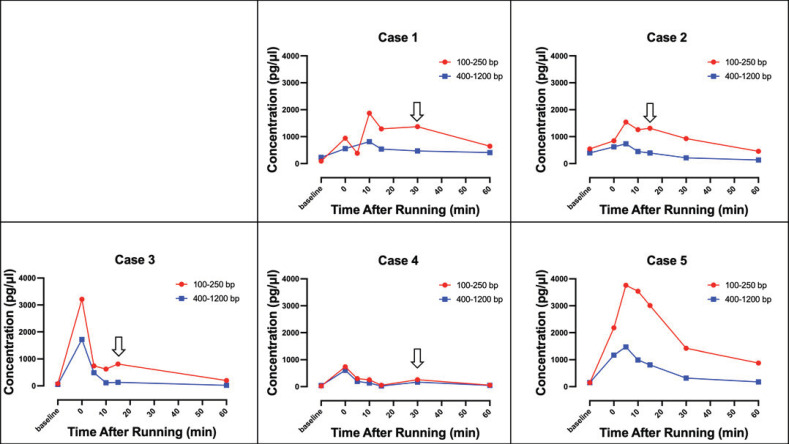
Temporal changes in plasma cell-free DNA (cfDNA) concentration following exercise. The concentrations of both 100–250 bp and 400–1200 bp cfDNA fragments were measured over time post-exercise. Concentration of cfDNA fragments of 100–250 bp is shown in red, while that of 400–1200 bp is shown in blue, with the time course represented in a line graph. The 100–250 bp fragments exhibited a significant increase immediately after exercise, followed by a gradual decline. Notably, in cases 1, 2, 3, and 4, a slight increase was observed between 15 and 30 min post-exercise, likely attributed to the degradation of the 400–1200 bp fragments, which resulted in a temporary rise in the levels of 100–250 bp fragments. The ⇩ symbol indicates the timing of this transient increase in the 100–250 bp cfDNA fragments.

**Table 1 diagnostics-15-00109-t001:** Participant demographic characteristics.

Case	Age (Years)	Body Weight (kg)	Body Height (cm)	BMI (kg/m^2^)
1	46	58	158	23.2
2	42	68	175	22.2
3	39	96	172	32.6
4	35	63	174	20.8
5	39	63	164	23.4
Mean	40 ± 4.1	70 ± 15	169 ± 7.3	24.4 ± 4.7

**Table 2 diagnostics-15-00109-t002:** Analysis of 100–250 bp cfDNA half-life.

Fragment Size (bp)	Case	Base (pg/μL)	Maximum (pg/μL)	Fold Increase (-Fold)	Half-Life (Minutes)
100–250	1	91.7	1870	20.3	32.7
	2	544	1540	2.83	31.4
	3	77.9	3210	41.2	14.9
	4	22.3	738	33.0	16.1
	5	146	3760	25.7	26.1
	Mean				24.2
400–1200	1	228	813	3.56	50.8
	2	397	736	1.85	22.4
	3	50.6	1720	33.9	9.5
	4	39.1	608	15.5	16.0
	5	149	1170	7.85	17.8
	Mean				23.3

**Table 3 diagnostics-15-00109-t003:** Half-life of cell-free DNA in literature search.

Study	Population	Sample Size	Sample Type	Collection Tube	Measurement Method	cfDNA Half-Life (Minutes)
Lo et al. [20] (1999)	Pregnant women	8	Venous plasma	EDTA tubes	Real-time PCR	16.3 (4–30)
Diehl et al. [21] (2008)	Cancer patients	1	Venous plasma	EDTA tubes	Real-time PCR	114
Breitbach et al. [22] (2014)	Healthy adults	13	Capillary plasma	EDTA tubes	Real-time PCR	15
Yao et al. [23] (2016)	Healthy adults	1	Serum samples	EDTA tubes	Real-time PCR	30.8

cfDNA: cell-free DNA, EDTA: ethylenediaminetetraacetic acid, PCR: polymerase chain reaction.

## Data Availability

The original contributions presented in this study are included in the article. Further inquiries can be directed to the corresponding author.

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
