# Peer review of "Dynamics and Half-Life of Cell-Free DNA After Exercise: Insights from a Fragment Size-Specific Measurement Approach"

_diagnostics, 2025, doi:10.3390/diagnostics15010109_

Round 1

Reviewer 1 Report

Comments and Suggestions for Authors

This exploratory study reports on a fragment size-specific evaluation of the half-life of cell-free DNA. As the first such study, it is of interest in spite of the small sample size. The study is overall well-written. 

Major comments:

- This study, in spite of the small sample size, as well as other studies reported in the literature search, suggests some level of inter-individual variability in the half-life of cfDNA, ranging between 15 and 30 minutes between different participants. This aspect should be addressed in the discussion

- In the discussion, the authors state that the half-life determined in their study at 24.2 minutes is shorter than in other studies. This statement is not clear given that two of the four studies listed in the literature search reported half-lives of 15 minutes.

- While the approach using Tape Station to assess fragment-size specific half-lives is interesting, the accuracy and technical variability of this quantification approach is not clear. For example, it is not clear if only a single measurement was performed per sample, or if technical replicates were performed to account for technical variability. 

Minor comments:

- Introduction, line 59: The statement on cfDNA degradation is not clear. Is this referring to cfDNA degradation after blood collection?

- Methods: The description of the extraction method is not clear as the first paragraph states that extraction was performed using the QuickGene cfDNA isolation kit, and the second paragraph states that extraction was performed using QuickGene-Mini8L.

Author Response

Response to Reviewer 1 Comments

We wish to express our appreciation to the Reviewer for his or her insightful comments, which have helped us significantly improve the paper.

Major comments 1: [This study, in spite of the small sample size, as well as other studies reported in the literature search, suggests some level of inter-individual variability in the half-life of cfDNA, ranging between 15 and 30 minutes between different participants. This aspect should be addressed in the discussion]

Response : [Thank you for your comment. We have added the sentences to the limitations section. “Previous studies have also reported variability in cfDNA half-life, and consistent with these findings, our study observed inter-individual differences. This suggests the need to evaluate factors such as age, renal and hepatic function, muscle mass, and nuclease activity to better understand cfDNA clearance mechanisms.”]

Major comments 2: [In the discussion, the authors state that the half-life determined in their study at 24.2 minutes is shorter than in other studies. This statement is not clear given that two of the four studies listed in the literature search reported half-lives of 15 minutes.]

Response : [Thank you for your comment. We have revised the text of discussion as follows.
"While the half-life of 24.2 minutes observed in this study falls within the range reported in previous studies (15–114 minutes), it is longer than the 15-minute half-lives reported in some studies.[20,22] We believe that the fragment size-specific measurement approach allowed us to capture cfDNA dynamics with greater precision."]

Major comments 3: [While the approach using Tape Station to assess fragment-size specific half-lives is interesting, the accuracy and technical variability of this quantification approach is not clear. For example, it is not clear if only a single measurement was performed per sample, or if technical replicates were performed to account for technical variability.­­­­­­­]

Response : [Thank you for your comment. We have added a description of intra-rater reliability to the methods and results.

" Materials and Methods

2.9. Intra-rater reliability

 The intra-rater reliability was assessed using the Intraclass Correlation Coefficient (ICC). ICC(1,1) was calculated for the single rater using the absolute agreement, one-way random-effects model. Python (version 3.11.8) was utilized for the analysis.

Results

3.2. Quantitative and qualitative analysis of cfDNA

Blood samples were collected at 0, 5, 10, 15, 30, and 60 minutes post-exercise, and the concentration and size of cfDNA were measured and analyzed using the 4150 TapeStation system. The analysis yielded an ICC(1,1) value of 0.943 (95% CI: 0.501–0.919), indicating good reliability. "]

Minor comments 1: [Introduction, line 59: The statement on cfDNA degradation is not clear. Is this referring to cfDNA degradation after blood collection?]

Response : [Thank you for your comment. We have revised the text of introduction as follows.

“First, they lacked measures to prevent cfDNA degradation by nucleases in the blood, particularly degradation caused by nuclease activity in the blood after collection.”]

Minor comments 2: [Methods: The description of the extraction method is not clear as the first paragraph states that extraction was performed using the QuickGene cfDNA isolation kit, and the second paragraph states that extraction was performed using QuickGene-Mini8L.]

Response : [Thank you for your comment. We have revised the text of “2.7. Isolation of cfDNA” as follows.

“The extraction of cfDNA was performed using the QuickGene cfDNA isolation kit (catalog no. CF-L; KURABO, Osaka, Japan) as the extraction reagent and the QuickGene-Mini8L (KURABO, Osaka, Japan) as the extraction device, following the manufacturer’s instructions.”]

Thank you once again for your valuable comments and suggestions.

Reviewer 2 Report

Comments and Suggestions for Authors

The manuscript is clear, relevant for the field and presented in a well-structured manner. The study was conducted very thoroughly.

But firstly, it is unclear why the authors limited themselves to a review for comparison of English-language studies only.

Secondly, there is too much subjunctive mood in the conclusion. It remains unclear whether the decay of longer fragments of extracellular DNA can really affect the jump in the level of shorter fragments?

In the introduction, it is worth mentioning for which diseases, besides oncology, extracellular DNA can be used as a biomarker.

Author Response

Response to Reviewer 2 Comments

We wish to express our appreciation to the Reviewer for his or her insightful comments, which have helped us significantly improve the paper.

Comments 1: [Firstly, it is unclear why the authors limited themselves to a review for comparison of English-language studies only.]

Response 1: [Thank you for your comment. In fact, no non-English articles were retrieved during our search; therefore, we have removed the statement indicating that the review was limited to English-language studies. I apologize for the misunderstanding caused by the incorrect description.]

Comments 2: [Secondly, there is too much subjunctive mood in the conclusion. It remains unclear whether the decay of longer fragments of extracellular DNA can really affect the jump in the level of shorter fragments?]

Response 2: [Thank you for your comment. We have revised the conclusion as follows.

" Conclusions:

Our study determined that the half-life of 100–250 bp cfDNA fragments is approximately 24.2 minutes, indicating rapid clearance. This rapid turnover highlights the suitability of cfDNA as a dynamic biomarker for monitoring disease progression and treatment response. This is the first study to specifically calculate the half-life of 100–250 bp cfDNA fragments using the PAXgene® Blood ccfDNA tubes and 4150 TapeStation system. The findings of this research will provide a scientific foundation for enhancing the clinical utility of cfDNA."]

Comments 3: [In the introduction, it is worth mentioning for which diseases, besides oncology, extracellular DNA can be used as a biomarker.]

Response 3: [Thank you for your comment. We have added the following statement. “Recent studies have increasingly explored the potential of cfDNA as a biomarker not only in oncology but also in autoimmune diseases, traumatic conditions, sepsis, cardiovascular diseases, and sports medicine.[11-15] ”]

Thank you once again for your valuable comments and suggestions.
